# Towards Understanding the Gene-Specific Roles of GATA Factors in Heart Development: Does GATA4 Lead the Way?

**DOI:** 10.3390/ijms23095255

**Published:** 2022-05-09

**Authors:** Boni A. Afouda

**Affiliations:** Institute of Medical Sciences, Foresterhill Health Campus, University of Aberdeen, Aberdeen AB25 2ZD, Scotland, UK; boni.afouda@abdn.ac.uk; Tel.: +44-1224-437490; Fax: +44-1224-437465

**Keywords:** transcription factors, heart development, *GATA4* genes, noncanonical Wnt signalling

## Abstract

Transcription factors play crucial roles in the regulation of heart induction, formation, growth and morphogenesis. Zinc finger GATA transcription factors are among the critical regulators of these processes. *GATA4*, *5* and *6* genes are expressed in a partially overlapping manner in developing hearts, and *GATA4* and *6* continue their expression in adult cardiac myocytes. Using different experimental models, GATA4, 5 and 6 were shown to work together not only to ensure specification of cardiac cells but also during subsequent heart development. The complex involvement of these related gene family members in those processes is demonstrated through the redundancy among them and crossregulation of each other. Our recent identification at the genome-wide level of genes specifically regulated by each of the three family members and our earlier discovery that *gata4* and *gata6* function upstream, while *gata5* functions downstream of noncanonical Wnt signalling during cardiac differentiation, clearly demonstrate the functional differences among the cardiogenic GATA factors. Such suspected functional differences are worth exploring more widely. It appears that in the past few years, significant advances have indeed been made in providing a deeper understanding of the mechanisms by which each of these molecules function during heart development. In this review, I will therefore discuss current evidence of the role of individual cardiogenic GATA factors in the process of heart development and emphasize the emerging central role of *GATA4.*

## 1. Introduction

Cardiovascular-related disease is a leading cause of death worldwide, and the prevalence of congenital heart defects is around 1% of live births, with up to 10% of spontaneously aborted foetuses found to be affected by cardiovascular diseases [1,2]. Thus, intense efforts have been made over the past few years to improve our understanding of the mechanisms underlying heart formation by characterising the molecular mechanisms involved. Amongst the first genes expressed during heart specification are those encoding a family of highly conserved zinc finger transcription factors named GATA factors [3] that have also been shown to be involved in the development of hematopoietic [4], digestive and reproductive systems [5]. Six paralogs were identified in vertebrates and originally divided fundamentally into a hematopoietic (*GATA1/2/3*) and a cardiac (*GATA4/5/6*) subfamily. The three cardiogenic GATA factors are known to be important and required for heart development as I discuss below (see Table 1 and Section 1). These three “sisters” (*GATA4*, *5* and *6*) have also been shown to be (to some extent) sufficient for promoting different aspects of heart development as their overexpression leads to the induction of cardiac differentiation. For instance, forced expression of Gata4 enhances cardiogenesis during Xenopus embryogenesis [6] or in P19 embryonal carcinoma cells [7]. In addition, Gata4 expression in stem cell-like embryonic explants was shown to be sufficient to induce cardiac differentiation of cardiomyocytes [8,9,10,11]. Studies using an engineered embryonic stem cell (ESC) line that allow conditional expression of *Gata*5 show that the expression of that single factor (and its sisters *Gata4* and *6* under defined culture conditions) is sufficient to promote abundant populations of beating and phenotypically normal cardiac cells [12]. Furthermore, although GATA4, 5 and 6 have all been shown to be implicated in heart formation, evidence about their specific roles in that process differs depending on the species in which the investigations were carried out. The apparent confusing picture that has then appeared due to these differences among species, makes identification of specific functions of specific cardiogenic GATA genes difficult to dissect. 

In our own research, in order to understand the specific function of individual cardiogenic *gata* during heart development, we profiled gene expression at a genome- wide level in settings where they were either individually depleted or depleted in combination and compared these to those in normal cardiac tissues [24,25] (Figure 1B). We were able to firstly identify genes specifically regulated by individual GATA factors during heart development in *Xenopus laevis* and secondly confirm that some are conserved in mammals. We also uncovered some functionally distinct roles for *gata4*, *5* and *6* during heart formation using the experimentally accessible Xenopus model. Wnt signalling pathways are involved in the processes that drive cardiac fate and development. Using antisense morpholino oligonucleotides to block the function of *gata4*, *5* and *6*, we showed that the loss of *gata4* and *6* results in the inhibition of both early and late cardiogenic markers, while the loss of *gata5* leads only to the inhibition of late cardiomyocyte differentiation markers [11] (Figure 1A). We demonstrated that while the effects resulting from the loss of *gata4* and *6* functions can be rescued by activating noncanonical Wnt11b signalling, the loss of *gata5* function cannot be rescued by noncanonical Wnt signalling, suggesting therefore that the differential requirements for GATA factors during cardiac differentiation are mediated by noncanonical Wnt11b signalling [11].

Another level of complexity in the functions of individual cardiogenic GATA factors in heart formation and development has emerged, firstly, from studies demonstrating redundancy [26,27] and crosstalk [28] between GATA factors during cardiogenesis (see in paragraphs below) and, secondly, from functional differences at cellular levels suggesting that some of these molecules drive cardiac fate not only cell-autonomously but also through mechanisms involving signalling molecules [9,11,28,29,30,31,32,33,34]. Thus, the above results obtained from a nonmammalian model system proved to be very informative and call for exploring further supportive information from mammalian model systems.

## 2. The Requirement of Cardiogenic *Gata* Genes for Heart Development

Indeed, implications of *Gata6* in heart formation originally emerged from the *Gata6* knockout mouse, which displayed extraembryonic endoderm defects leading to early death prior to heart induction [35,36]. It has also been shown that elevated levels of Gata6 delay the onset of maturation of cardiac precursors [37], but as I will discuss below, increased levels of *Gata6* may compensate for the loss of *Gata4* [14,22,23,38,39]. Studies in Xenopus [11] as well as in zebrafish have confirmed a requirement of *gata6* in cardiogenesis, which, however, appears to be at the later stage of maturation (that is cardiac differentiation) rather than initial induction [30]. Further evidence has emerged from genetic studies that implicated *Gata6* not only in cardiomyopathies but also in phenotypes leading to lethality [16,17,18,40,41,42].

For *Gata5*, in vitro studies revealed its requirement for differentiation of committed cardiogenic precursors into endocardial cells (Ecs) [19]. In zebrafish, important roles of *gata5* in heart formation have been shown in studies with the *gata5* (faust) mutant that lack endocardial cells resulting in a reduced number of cardiomyocytes [43]. Furthermore, targeted inactivation of *Gata5* in mice [44] has shown the importance of *Gata5* in the aetiology of the bicuspid aortic valve (BAV) defect during heart development [44]. In that study, the deletion of *Gata5* specifically from endocardial cells was sufficient to recapitulate the cardiac phenotype of *Gata5* knockout mice, suggesting a cell-autonomous function of *Gata5* in regulating endocardial cushion differentiation and the importance of that factor in mammalian heart development and congenital heart disease [45,46]. As I will discuss in more detail below, *Gata4*/*5* compound mutant mice have been described to show more severe cardiac defects [47], and *Gata4*/*5* and *Gata5*/*6* compound heterozygotes present double outlet right ventricle and ventricle septal defects [15]. Although those studies highlight the importance of *Gata5* in heart development, they also indicate a possible redundancy in function of GATA factors in heart formation. In that regard, studies in zebrafish and Xenopus have shown that *gata5* has redundant roles with *gata6* in cardiac progenitor specification [16,45,46].

With regards to *GATA4*, mutations in the *GATA4* gene were shown to cause congenital cardiomyopathies [48,49,50,51,52,53,54,55,56,57,58,59]. However, in mice, although *Gata4* knockout was shown to be embryonically lethal, its heart phenotype was shown to be secondary due to defects in the extraembryonic endoderm, with fusion at the ventral midline not occurring, associated with a cardia bifida phenotype [22,23]. Likewise, the depletion of *gata4* in zebrafish has also produced morphological defects [60]. In tetraploid compensation experiments to circumvent the extraembryonic defects in *Gata4*-null mice, the derived embryos lacked proepicardium and showed defects in cardiac morphogenesis that included hypoplastic ventricular myocardium [61]. However, as I will discuss below, redundant roles for both *Gata4* and *6* in controlling the onset of cardiac myocyte differentiation have been shown in recent studies in which double *Gata4*/*6* KO embryos completely lack hearts, although second heart field progenitor cells are still generated [62]. In addition to its requirement for epicardial and myocardial formation, *Gata4* has also been shown to be involved in endocardial cells and valve formation [63,64]. Indeed, in studies assessing the transcriptional mechanism underlying valve development in mice, Stefanovic et al. have shown that *Gata4* is involved in the activation and repression of atrioventricular-specific genes by acetylation and deacetylation of the atrioventricular (AV) canal gene loci [64]. This identifies GATA binding elements as a potential platform that recruit broadly active histone modification enzymes and cell-type-specific cofactors in order to drive cell-type-specific gene expression programmes. Indeed, other studies have provided evidence that the cooperation of *Gata4* with active histone modification enzymes and localized cofactors ensured a proper spatial–temporal expression of cardiac genes during cardiac differentiation [65,66,67,68]. Although the above-described studies have provided a direct requirement for *Gata4* in cardiomyocyte specification, in addition, gain-of-function studies suggest that *Gata4* is sufficient to affect cardiomyocyte fate. For instance, forced expression of Gata4 enhances cardiogenesis during Xenopus embryogenesis [6] or in P19 embryonal carcinoma cells [7]. Furthermore, Gata4 expression in stem cell-like embryonic explants was shown to be sufficient to induce differentiation of cardiomyocytes [8,9,10,11]. Those data suggest that *gata4* is a vital candidate for enhancing cardiogenesis from a progenitor cell population.

Altogether, those experiments with both those obtained based on its expression [3,5] and the phenotypes generated by various gain-and loss-of-function experiments suggest that *Gata4* could be viewed as a key GATA regulator of cardiogenesis as discussed below. 

The current review therefore identifies important themes emerging from recent advances made in understanding the role of cardiogenic GATA factors in heart formation with a justification to focus on *Gata4*.

## 3. Redundancy and Functional Interactions among Cardiogenic *GATA* Genes

Significant efforts made over the past to elucidate the molecular mechanisms involved in heart disease have identified some genes whose functions are linked to congenital heart disease (CHD) [69,70,71]. The emerging picture from combined human and mouse genetic studies is that mutations in different genes (such as *GATA*, *NKX2.5* and *TBX5*) can lead to similar cardiac defects (polygenic), while mutations in the same gene can often lead to varying defects, for instance, in different settings (pleiotropic), suggesting a certain complexity in CHD. Such complexity for the mechanisms of CHD is also evident at both genetic and cellular levels, as multiple cell lineages contribute to proper heart formation. Several lines of evidence confirm that CHD is heritable and that *GATA4*–*6* are among the genes that are linked to CHDs in humans [16,17,18,20,40,45,46,48,56,58,59,70,72,73,74,75,76,77]. For example, mutations in *GATA4* have been associated with atrial and/or ventricular septal defects, Tetralogy of Fallot (TOF) and persistent truncus arteriosus (PTA) [47,48,56,78] (also see Table 1). Interestingly, heterozygous mutations of *Gata4* in mice recapitulate the human phenotype [49]. However, a level of complexity regarding the involvement, functions and interactions of GATA factors in CHD have been shown in studies in which the loss of both *Gata4* and *Gata6* in mice leads to acardia, suggesting that genetic interactions between these factors are essential for the onset and/or maintenance of cardiogenesis [62,78]. In the former study, in vitro investigations using either *Gata4* or *Gata6* or a combined deletion of both factors in ESC, as well as embryos generated from those, confirm functional interactions between those two factors. When either *Gata4* or *Gata6* was disrupted, the number of beating embryoid bodies was dramatically reduced to a similar level. That observed effect for *Gata4* is similar to that previously reported for the loss of *Gata4* in ES embryoid bodies and may reflect a loss of extraembryonic endoderm necessary for maturation, rather than a cell-autonomous effect [79]. Moreover, a more penetrant effect was observed when one allele of *Gata6* was disrupted in the *Gata4*^−/−^ ES cells, with a complete lack of cardiac differentiation in *Gata6*^−/−^. Since no effect on cardiac progenitor status was observed, those studies confirm that *Gata4* and *Gata6* have some redundant functions in controlling cardiac myocyte differentiation and furthermore suggest that they are dispensable for the formation of cardiac progenitor cells. Those observed functional interactions between *Gata4* and *Gata6* were confirmed in the heart of embryos generated from *Gata4*^−/−^ ES cells by tetraploid complementation [79]. Embryos with acardia or a complete lack of hearts observed from this experimental setting confirmed such functional interactions between *Gata4* and *Gata6* in vivo. In those studies, the absence of the heart in *Gata4*/*6*-null embryos was proven not to be a consequence of a general embryonic arrest or a defect in gastrulation, as the *Gata4*^−/−^*Gata6*^−/−^ embryos did retain many features characteristic of the developmental stage E8.5 where the investigations were carried out [62]. Those observations are in line with previous studies in which it was shown that mice embryos with compound heterozygous mutations in *Gata4* and *Gata6* die at the embryonic stage around E13.5 due to several cardiac defects, among which one can cite vascular defects, PTA, evidence of failed outflow tract (OFT) septation as well as myocardial thinning, indicative of functional interaction between *Gata4* and *Gata6* in cardiac and vascular development [80,81]. Thus, those additional studies have recognised that genetic redundancy exists between *Gata4* and *Gata6* and that coexpression of these transcription factors characterises functional compensation [80]. Subsequent studies were conducted to shed light on the genetic redundancy between the GATA factor function during cardiogenesis [27,82]. Evidently such investigations needed to overcome the challenge of avoiding interfering in other developmental processes in which those factors are involved. Since zebrafish embryos do not rely on the equivalence of mammalian extraembryonic endoderm for early development, they provide an excellent model to generate triple *gata4*/*5*/*6* knockout to probe genetic interaction among these genes. In the study by Sam et al. [27], the authors generated multiple combinations of double and triple *gata4*/*5*/*6* heterozygous and homozygous mutant offspring to define the individual and redundant functions of *gata4*/*5*/*6* during different stages of embryonic heart development. The study not only describes individual roles of GATA factors but also describes evidence for genetic compensation and dosage requirements for *gata4*/*5*/*6*-depedent cardiogenesis [27]. 

Another level of complexity in the interaction among the three cardiogenic *Gata* sisters has been shown in other investigations that involved *Gata5*. Laforest and Nemer [15] investigated the genetic interaction between *Gata4*/*Gata5* and *Gata5*/*Gata6* in developing mice embryos. In that study, very high lethality was observed in *Gata4*^+/−^*Gata5*^+/−^ double heterozygote embryos, which seems not to be due to retarded development but rather displayed defects identified as double outlet right ventricle (DORV) and ventricular septal defect (VSD) [15]. As both GATA4 and GATA5 are expressed in the atrioventricular cushion, analysed embryos show that the observed defects arose from the abnormal development of endocardial cells, suggesting genetic cooperation between *Gata4* and *Gata5* in endocardial cushion formation. Interesting, the few *Gata4*^+/−^*Gata5*^+/−^ double heterozygotes that survived to adulthood presented with a massively hypertrophied aortic valve, consistent with defective valve development leading to aortic stenosis. Those data are in line with those from previous studies indicating that *Gata4* and *5* cooperatively regulate cardiac myocyte proliferation in mice [47]. The study provided evidence that although *Gata5*^(−/−)^ mice are viable, *Gata4*^(+/−)^ and *Gata5*^(−/−)^ mutants die at midgestation and exhibit profound cardiovascular defects, including abnormalities of cardiomyocyte proliferation and cardiac chamber maturation. These results demonstrate functional redundancy between *Gata4* and *Gata5* during cardiac development. Similar to the *Gata4*/*Gata5* compound discussed above, genetic interaction between *Gata5* and *Gata6* also shows a very high embryonic or perinatal lethality as revealed in *Gata5*/*Gata6* heterozygote compounds [15]. The aetiology of those lethalities shows the presence of DORV and VSD but interestingly only at the level of the aortic valve. The results from the above-mentioned studies clearly show an interesting and complex cooperative genetic interaction revealed in these *Gata4*/*Gata6*, *Gata4*/*Gata5* and *Gata5*/*Gata6* compounds, resulting in similarity with some subtle distinctions at the level of phenotypes, as well as affected molecular gene targets. Such complexity is also evident at the cellular level as multiple lineages contribute to proper heart development. Thus, although it appears that deciphering the genetic programme controlled by individual cardiogenic GATA factors is challenging, attempts have been made at elucidating individual contributions of these factors in the complex regulatory network that governs proper heart formation.

The above evidence mainly from mouse studies and to some extent zebrafish studies [27] strongly suggests that GATA factors may have redundant activities during cardiogenesis, a redundancy that is more likely as they all bind to the same binding sites with high affinity [83,84]. In addition to this, the peculiarities of the species in which the investigations were carried out make deciphering their redundancy difficult. The apparent confusing picture provided the rationale behind other investigations that aimed to improve our understanding of the redundancy function of GATA factors during cardiogenesis. This is exemplified in Peterkin et al. [26] in which the authors set out to knock down each of the cardiogenic GATA factors individually or in combination using antisense morpholinos in the experimentally accessible Xenopus and zebrafish models. In those studies, the knockdown of *gata4* or *gata5* individually had little effect on myocardium formation, but a clear phenotype was observed when both were knocked down together in Xenopus. In contrast, the knockdown of *gata5* on its own in zebrafish had a major effect [26,43]. However, in both species, *gata6* knockdown had a clear effect that was further enhanced by knocking down either *gata4* or *gata5* in the same embryos. This observed and more important role that *gata5* plays in myocardium formation in zebrafish highlights changes in the use of the GATA factors during evolution. The relatively earlier onset of expression of Gata5 in zebrafish has been provided as a possible explanation. In other words, Gata5 is expressed in zebrafish before Nkx2.5 (a critical cardiac specifying gene), which is known to be active before GATA factors in Xenopus and in mammals. This supports the finding that these factors regulate each other in both Xenopus and zebrafish and more importantly suggests that the establishment of such mutually supportive circuits is more important than the particular order in which the two genes become active. Such a mutually supportive circuit is also observed with bone morphogenic protein (BMP), a key signalling molecule in cardiac differentiation. Interestingly, noticeable differences appear between Xenopus and zebrafish with regards to BMP signalling where nkx is active before bmp signalling in Xenopus in contrast to zebrafish where bmp signalling is active before Gata or nkx. A similar circuit is observed in invertebrates, namely Drosophila, where BMP is known as DPP [26,85], supporting the concept of some vital core regulatory circuits named kernels that operate in organogenesis during embryonic development [86]. 

Taken this functional redundancy into account, and as I will discuss below the mutual regulation between cardiogenic Gata genes, we can now attempt to deconvo-lute, distil and dissect the specific functions of GATA6, 5 and 4 in heart formation.

## 4. Direct and Indirect Roles of *Gata6* in Cardiac Development: Is *Gata6* the Junior Partner?

Mammalian *Gata6*-null mice die prior to heart induction at E5.5–7.5 (see Table 1) due to defects in extraembryonic endoderm [35,36] consistent with *Gata6* function early in development. In mice, chimera embryo studies have shown that *Gata6*^−/−^ ES cells were able to contribute to heart and differentiate into myocardium in vitro, suggesting a non-cell-autonomous role of *Gata6* in heart defects. Gain-of-function studies in Xenopus, however, have suggested a role of *gata6* in cardiogenesis [37]. In that study, it has been shown that proper heart development requires a decrease in expression of Gata6 in cardiac precursors, as elevating its expression beyond this time-delayed cardiomyocyte maturation. Although these findings suggest that Gata6 may hold cardiac precursor cells in a progenitor and proliferative state, further loss of function in both Xenopus and zebrafish have suggested instead that the *gata6* function is required for the maturation of cardiac progenitors rather than for their initial induction [30,32]. Such function for *gata6* in the developing myocardium was shown to be dependent on Bmp4 expression [29,30]. Such involvement of signalling molecules downstream of *gata6*, and so its cell nonautonomous function, was subsequently confirmed in Xenopus where we have shown that *wnt11b* mediates the differential requirements of GATA factors during cardiogenesis [11]. In that study, we identified that during the process of heart formation noncanonical Wnt signalling mediated by *wnt11b* functions downstream of *gata6* (and *gata4* but not *gata5*) (Figure 1A). 

In humans, mutations in *GATA6* have been identified in CHD highlighting its crucial role in heart formation [87] (also see Table 1). Those mutations are associated with structural and conduction system abnormalities that include valve (BAV) and septal defects [32,34,88]. In recent years, more investigations have been carried out that confirm the implication of *GATA6* in OFT [16,17,18]. To illuminate the molecular mechanisms for those cardiovascular malformations, recent studies in human-induced pluripotent cells (hiPSCs) show that *GATA6* regulates *SMYD1* that activates *HAND2* and that *KDR* with *HAND2* orchestrates OFT formation [89]. Although the above-mentioned studies show *GATA6*’s role in CHD, *GATA6*, like its sister *GATA4*, has also been shown to be involved in postnatal cardiac regulation, for instance, in cardiac hypertrophy [41] (also see Table 1) and haemostasis. It has been shown that its hypertrophic role requires *GATA6* interaction with a small heterodimer partner (a repressor) [88] and, interestingly, with its sister *GATA4* [90]. Those data suggest the importance of interacting partners and consequently post-translational modifications in *Gata6* (*Gata5* and *Gata4* as detailed below in Section 5 and Section 6) cardiac function. Among those post-translational modifications, the s290 phosphorylation [91,92] has been reported to be required for its role in vascular smooth muscle formation and its SUMOylation [93], and possibly its epigenetic modifications are involved for its role in heart development such as for its sister *Gata4* [66] (as discussed below).

A further level of complexity regarding the function of *Gata6* has been revealed in subsequent ES cell studies. In those studies where conditional expression of *Gata6* in a mouse ES cell line was generated, experimental expression of *Gata6* was shown to be able to induce cardiac differentiation in a serum-free medium in a similar way as *Gata4* and *Gata5*. However, some differences did appear among the cardiogenic GATA factors in those assays. Among those, *Gata6* was less efficient at inducing cardiac differentiation (compared to its other sisters), and there were notable differences with regards to cardiac markers induced by each of them [12]. It has been suggested that the differences might reflect some early crosstalk among the genes themselves as they can activate expression of each other to different extents, with *Gata6* specifically not able to activate the expression of *Gata5*. Among the possible explanations regarding the differences in genes activated by experimental *Gata6* expression, it was suggested that some distinct cardiac cell types were generated in this *Gata6*-programmed culture [94]. These observations highlight once more our need for the identification of specific target genes for those factors in order to improve our knowledge of the gene regulatory network (GRN) in which they are involved during cardiogenesis. To this end, we have recently undertaken work aiming to identify *gata6* transcriptionally regulated genes (as well as those of its other sisters *gata4* and *gata5*) on a genome-wide scale through RNA-seq analyses [24,25]. The data generated in that study provide a useful platform for future investigations that will further contribute to elucidating molecular pathways and the GRN underpinning embryonic cardiogenesis.

Other insights into *GATA6* functions in cardiac differentiation came from recent work performed on long noncoding RNAs (lncRNAs) [95]. It was found that in humans, coding *GATA6* resides adjacent to its antisense counterpart lnc*GATA6* (*GATA6-AS1*) on the genome, with both sharing the same promoter acting bidirectionally. Interestingly, it was found that *GATA6-AS1* regulates the expression of its coding counterpart (*GATA6*) and thereby cardiomyocyte differentiation.

## 5. Direct and Indirect Roles of *gata5* in Cardiac Development: Listen to the Fish!

In zebrafish, *gata5*-null embryos show a cardia bifida phenotype that resembles the mouse *Gata4*-null embryos. However, in mice, *Gata5* homozygous embryos show no obvious phenotypes. Although this might indicate that functions may have been differentially assigned to these different GATA genes during evolution, it is worth noting that *Gata5* mutant mouse embryos still express a truncated but potentially transcriptionally active isoform of *Gata5* [96]. This is consistent with the observation of a shared characteristic among members of the GATA family to possess several alternative promoter-first exon combinations. In fact, in humans and mice, *GATA6* possesses two distinct promoter-first exon combinations [97], whereas in chickens, two isoforms of *GATA5* are produced [98]; one isoform excludes exon 2 (which was targeted in *Gata5* mutant mice) and produces a truncated yet active protein, indicating that the observed phenotype in *Gata5* mutant mice could potentially not reflect the absence of the full role of *Gata5* in mouse cardiogenesis. Clear evidence, however, of an essential role of *gata*5 during cardiogenesis in zebrafish has emerged [43]. In fact, in contrast to *Gata4*-null mice, which seem to still express terminal differentiation markers, in zebrafish, mutant early markers (*nkx2.5*) and terminal differentiation markers are reduced [43]. That essential role of *gata5* in heart formation as shown in zebrafish is further supported in studies in which it has demonstrated that *smarcd3b* cooperatively with *gata5* promotes cardiovascular progenitor cell fate [99]. As mentioned above for *Gata6*, those data suggest the importance of interacting partners and consequently post-translational modifications in *gata5* cardiac function. In line with this, it has been demonstrated that the SUMOylation of Gata5 is indispensable for its function in zebrafish cardiac development [100]. Yet, a conserved role of *gata5* across species has emerged, as faust (*gata5* mutants in zebrafish) did display endocardial defects [43], a phenotype also observed in mice [19]. These observations were further confirmed in recent studies in which a prominent role of mouse *Gata5* in valve formation was described [15,44]. Valve development is a complex process that involves the expansion and differentiation of endocardial cells and their migration after epithelial-to-mesenchymal transformation (EMT) to form endocardial cushions at the atrioventricular canal (AVC) and within the outflow tract (OFT). A proper formation of the OFT requires not only functions of “cardiogenic” GATA factors but also a significant role from a member of the “haematopoietic” GATA group, namely *GATA3*. This is revealed in a study where *Gata3* lethality was attenuated by catecholamine-based rescue [101]. The defects present in such mutants include a shorter OFT and reduced rotation of truncus arteriosus during the looping stages. Interestingly, individuals with DiGeorge syndrome with deletion in the 10p region that contains the *GATA3* gene present with multiple problems, including cardiac defects [102,103]. However, *Gata5*, a member of the “cardiogenic” subfamily, is expressed in the myocardium, as well as the endocardium and derived endocardial cushions. Conditional inactivation of *Gata5* in endocardial cells leads to hypoplastic hearts and a partially penetrant bicuspid aortic valve (valve with two leaflets rather than the normal three) [44] suggesting an autonomous role of *Gata5* in endocardial cushion formation; as discussed above, cardiac valve formation and compound mutants *Gata5*/*Gata4* and *Gata5*/*Gata6* have various defects in OFT in mice [15]. The above-mentioned data clearly support the importance of *Gata5* in valve formation, and they also confirmed more recent studies in which the *Gata5* loss-of-function mutation were found to be associated with congenital aortic valve [46] and Tetralogy of Fallot [45] (also see Table 1).

Recently it has also been demonstrated that *GATA5* plays a significant role in endothelial homeostasis and blood pressure [20]. Keeping in mind the role of blood pressure in heart remodelling, those data implicate *GATA5* as a major regulator of adult heart function. More recent investigations have provided the molecular mechanisms involved in *Gata5*’s remodelling function of the adult heart. They identified that the upstream regulation of *Gata5* in that process is linked to activation by *Sirt6*, which results in the downregulation of the *Gata5* repressor *Nkx2.3* [104]. 

While those former observations are again indicative of genetic interactions among all three cardiogenic GATA factors in heart defects, attempts to clarify mechanisms by which *gata5* functions during cardiogenesis have been made in studies using Xenopus as a model system [11,105]. Interestingly, the *gata5* function has been shown to be downstream of its sisters *gata4* and *gata6* to allow differentiation of heart precursor cells to mature cardiomyocytes [11] (Figure 1A).

Further evidence for an essential and direct role of *Gata5* in the process of heart formation has been provided in studies in which the conditional expression of *Gata5* in the ES cell line was used. The cells have proven to be capable of promoting cardiogenesis in a serum-free culture condition [12]. In contrast to what was observed in Xenopus where *gata5* function seems to favour subsequent differentiation of cardiac precursor cells [11] (Figure 1A), this mES cell context rather argues that *Gata5* induction directs early mesoderm-committed progenitors to cardiac fate [12]. Here, it has been shown that *Gata5* induction directs the development of cardiomyocytes as monitored by the phenotypical observation of beating sheets of cells that express cardiac differentiation markers and, more importantly, show a full range of action potential morphologies that are responsive to pharmacological stimulation [12]. Such a discovery provides scope for potential applications for drugs testing. These observations in mESCs that *Gata5* is more potent when compared to its other sisters, *Gata4* and *Gata6*, highlight once more our need for the identification of specific target genes for each of the cardiogenic GATA factors. In this regard, recent studies in *Xenopus laevis* were carried out by us to identify *gata5*-regulated genes on a genome-wide scale through RNA-seq analysis [24,25]. As mentioned for *gata6* above, the data generated in that study provide a useful platform for future investigations that will further contribute to elucidating molecular pathways and the GRN that governs heart formation.

## 6. Direct and Indirect Roles of *GATA4* in Cardiac Development: Is *GATA4* the Boss?

*GATA4* appears to be the most-studied of the three cardiogenic GATA factors, indicative of its role in cardiogenesis and possibly the complexity of its functions in that process. GATA4 is expressed throughout development and also in the adult heart. Clinical studies have linked its role in both human CHD, specifically atrial/ventricular septal defects [48,53,56,57,106,107] in heart defects in mice [47,48,78] (also see Table 1). Those defects were shown to result from *Gata4* alone or from a combination with its other sisters [15,62]. However, some differences have been observed regarding the differences in the severity of the phenotypes depending on the species and methods used in the studies. In studies using P19, a cell line induced to form beating cardiomyocytes by the addition of DMSO, the depletion of GATA4 by the antisense strategy prevented terminal differentiation and promoted apoptosis, while GATA4 gain-of-function studies suggest ectopic beating cardiomyocytes [108] in the absence of DMSO [7,106]. Consistent with this, gain-of-function experiments in Xenopus embryos cause a premature expression of cardiac differentiation markers and induce heart marker expression in the presumptive ectoderm and subsequently spontaneous cardiomyocyte beating [6,8,9,11,13]. However, most *Gata4*-null mice survive gastrulation and form differentiated myocardium, but show a ventral morphogenesis defect resulting in cardia bifida [22,23]. Those in vivo studies have shown that *Gata4* is crucial for proper heart tube formation, ventral patterning and proper heart assembly [22,23]. The observed cardiac defects in *Gata4*-null mice are supposed to be non-cell-autonomous, as *Gata4^−/−^* ES cells can contribute to an apparently normal heart in chimeric mice [23,79]. Chimera studies also suggested that the ventral phenotype observed in *Gata4*^−/−^-null mice is due to lack of GATA4 in the endoderm [107] rather than in the mesoderm, indicative of *Gata4* function involving downstream secreted factors emanating from the endoderm, which then induce cardiac differentiation in the adjacent mesoderm. That observation was confirmed in later studies in Xenopus where *gata4* was shown to induce cardiac differentiation through Wnt signalling [9,11] (also Figure 1A). In those studies, it was shown that β-catenin-dependent Wnt signalling restricts cardiogenesis via the inhibition of *gata4* (and to some extent *gata6* [32]) gene expression, where experimentally reinstating *gata4* function overrides β-catenin-mediated inhibition and restores cardiogenesis [9,32]. In turn, *gata4* directly regulates *wnt11b* gene expression, which is required to a significant degree for mediating the cardiogenesis-promoting function of *gata4* [9] (also see Figure 1A). In line with those data implicating the endoderm as a source of secreted factors for inducing cardiogenesis in the mesoderm, embryonic explant assays have implicated anterior endoderm as a tissue from which cardiac inductive cues are derived, including members of the *Nodal*, *Bmp*, *Fgf* and Wnt signalling pathways [109]. Explant assays in Xenopus identified *dkk1* (a Wnt inhibitor) as a secreted factor required for the expression of *hex1* in the endoderm associated with presumptive cardiac mesoderm [110,111], again supporting the rationale behind endoderm-derived factors inducing cardiac fate in mesodermal cells. It is tempting to suggest an attractive model in which *Gata4* plays a role in the inductive signals emanating from the endoderm to the adjacent mesoderm (Figure 2). Indeed, data from ES cells support such an assertion. Thus, as a direct overexpression of *Gata4* in murine ES cells causes them to differentiate into the extraembryonic endoderm [112], a system to conditionally express *Gata4* in cultured embryoid bodies (EBs) has been generated to evaluate the ability of *Gata4* to direct cardiomyocyte fate [31]. In such a setting, the induction of *Gata4* significantly increases the levels of *Sox17* expression, a known endoderm specification gene [113], as well as *Hex* expression. In the above-mentioned studies where conditional *Gata4* expressing cells were used, it appears that *Gata4* acts upstream of *Sox17* for the production of endoderm-derived heart-inducing factors [31]. Those data are consistent with previous observations in which the inhibition of *Sox17* with RNA interference resulted in the suppression of cardiomyocyte differentiation, without altering endoderm formation [114]. More importantly, these authors have shown that *Sox17* induces cardiomyogenesis in a non-cell-autonomous manner through the endodermal cardiogenic factor *Hex* [114] consistent with previous Xenopus studies [110,115].

### 6.1. Co-Operative Activity with Gata4 in Cardiogenesis: With a Little Help from My Friends

While those mixed EB culture conditions have shown that the *Gata4* function is to produce a cardiac-inducing endodermal signal, forced expression of *Gata4* directly in embryonic mesoderm has proven not to be sufficient to induce cardiac fate [116]. Instead, *Gata4* does promote cardiac specification and differentiation when coexpressed in noncardiac mesoderm with *Tbx5* (Figure 3) and *Baf60c* [116]. The co-operative activity among *Gata4* and other cardiac-specific transcription factor(s) for reprogramming various cell types into beating cardiomyocytes was confirmed in further studies but interestingly with different efficiencies [68,75,117,118,119,120]. Thus, it was reported that in vitro a minimum mixture of *Gata4*-*Mef2c*-*Tbx5* (GMT) (Figure 3) directly induces cardiomyocyte-like cells (iCMs) from mouse fibroblasts [121]. Subsequently, other groups also reported the generation of functional cardiomyocytes from mouse fibroblasts with various combinations of transcription factors that include either GMT plus *Hand2* (GHMT) (Figure 3) or *Mef2c*, Myocardin (*Myoc*) and *Tbx5* or using microRNAs [122,123,124]. Although in those in vitro settings full reprogramming into beating cardiomyocytes was shown not to be efficient [75,125], the gene transfer of GMT or GHMT cells into mouse hearts generated new cardiomyocytes from endogenous cardiac fibroblasts and improved cardiac function after myocardial infarction [68,73,74]. Interestingly, attempts to generate cardiomyocytes directly from postnatal human fibroblasts have also been made. In those studies, it was found that GMT was not sufficient for cardiac reprogramming, yet the addition of *MESP1* and *MYOC* was able to generate cardiomyocyte-like cells [126]. Although those induced cardiomyocytes (iCMs) express a panel of cardiac-specific genes, they seem to represent relatively immature cardiomyocytes as indicated by their cell morphology, expression of the embryonic cardiomyocyte marker α-smooth muscle actin (α-SMA) and slow Ca^2+^ oscillations. Because those iCMs require a coculture with murine cardiomyocytes to differentiate into beating cardiomyocytes, it was suggested that they might be similar to the early stage of embryonic cardiomyocytes before the start of contraction or alternatively that they might be some partially reprogrammed cardiomyocytes at a preinduced pluripotent (pre-iPSC) stage that can become fully pluripotent with additional stimuli (ibid). In this regard, it is worth noting that the stoichiometry of the transcription factors is believed to play a crucial role in those settings for determining the cell fate of the transduced cells into iCMs [120] or into smooth muscle-like cells. As the addition of *MYOC* upregulates sarcomeric genes and *MESP1* induces intracellular Ca^2+^ oscillations [126], it was suggested that *MYOC* and *MESP1* differentially regulate cardiac gene expression in human cardiac reprogramming (ibid). Although several other factors (for instance the gene family of Friends of GATA—also known as FOG) have been described to interact with GATA4, it is expected that the identification of the others will surely follow as further progress is made towards deciphering the GRN involved in driving cardiogenesis. A body of work confirms the importance of the FOG in regulating GATA factors functioning during cardiac differentiation and also provides the molecular mechanisms involved in their interactions with GATA4 [127,128,129,130] (also see Figure 3). Other studies identified ART27 as a GATA4 corepressor interacting with the Friends of GATA2 (FOG2) and NKX2.5, involved in downregulating cardiac-specific genes [131]. Whatever mechanisms are involved in the regulation of *GATA4* function during cardiac differentiation, these data clearly highlighted the crucial role played by *GATA4* in cardiomyocyte development.

### 6.2. Gata4 Possesses Intrinsic Cardiogenic Activities

Although the above-mentioned evidence established a co-operative activity of GATA4 with other factors in cardiogenesis, GATA4’s ability to programme (on its own) cardiac fate cannot be ruled out. Indeed, through the conditional expression of *Gata4* in a mouse embryonic stem cell line, *Gata4* was shown to be sufficient to highly expand cardiac cells [12], as monitored at the phenotypical level as well as transcriptomics followed by gene ontology showing a high enrichment of a cardiac muscle gene set [12]. Further efforts have been made to shed light on the cardiogenic activity of GATA4 by dissecting the role played by different domains that make up the GATA4 protein. It turns out that its cardiogenic activity requires a sequence stretch of 24 amino acids (129 to 152) that is also needed for GATA4’s interaction with the chromatin remodelling protein BAF60c [132]. As the same domain was shown not to be essential for its endodermal induction activity, this suggests that it acts as a cell-type-specific transcriptional activation domain. The serine at position 105, which is known to be phosphorylated by MAP kinase [133] and required for myocyte survival and hypertrophy [133], has been shown to be dispensable for the GATA4 induction of cardiogenesis. That same serine residue has been shown to be required for GATA4 to synergise with SRF but interestingly not with other important cardiogenic transcription factors, such as TBX5 and NKX2.5 [132]. Furthermore, it was demonstrated that the carboxy terminal region, composed of amino acid 362–400, physically and functionally interacts with CDK4, resulting in the enhancement of the cardiogenic activity of GATA4 [134]. Those data confirm the relationship between GATA4 and cyclin molecules as described in previous studies in which cyclin molecules were shown to be *Gata4* targets [135,136].

The above-mentioned data clearly indicate the importance of phosphorylation in the regulation of GATA4 cardiac function (Figure 3). They confirm previous studies that demonstrated GATA4 is phosphorylated by PKC at serine 419/420, allowing it to interact with STAT1 and transactivate the *Nppa* promoter [136]. Those post-translational modifications were reported to be crucial for the cardiac hypertrophic function of *Gata4*. Thus, over the past several years, the molecular mechanisms behind cardiac hypertrophy have been investigated [34,137,138,139,140]. More evidence for the implication of GATA4 phosphorylation in cardiac diseases was provided in another study [67]. Here, the authors demonstrated that the repression of GATA4 caused by its phosphorylation by beta-catenin leads to some observed diseases in humans and mice (ibid). These data are promising for the development of potential treatment for heart failure therapy and for heart repair after injury.

### 6.3. Context-Dependent Activities of GATA4

As just mentioned above with GATA4 phosphorylation at various sites (Figure 3), post-translational modifications account for many GATA4 activities in heart formation. The same thing was confirmed in other studies [141]. Another level of complexity in the mechanisms by which *Gata4* exerts its activities has been provided by studies in which a context-dependent regulation of *Gata4* was shown to take place via tuning its functions to avoid its dysfunction that would otherwise result in some heart diseases, such as arrhythmias [142]. While the initial heart tube is formed and elongated, specific regions differentiate and start to proliferate to form expanding atria and ventricle chambers that are divided by the atrioventricular canal. An important study was able to demonstrate that *Gata4* plays a crucial role in allowing those chambers to keep their identities for proper heart development [64]. The authors assessed the genome-wide distribution of H3K27ac by ChiP-seq. Although the genome contains thousands of occupied GATA sites, the question of why only a fraction of these sites have AV canal-specific properties was evident. The investigation found that GATA-binding elements in AV canal-specific enhancers recruit a *Gata4*/*Smad4*/Histone acetylase transcriptional activation complex in the AV canal and a *Gata4*/*Hey1*,2/Histone deacetylase transcriptional repression complex in the chambers [64]. The transcriptional activation complex induces H3K27 acetylation that is associated with AV canal gene activation, and the repression complex induces H3K27 deacetylation associated with the repression of AV canal genes in the chamber [64,66]. Those data that demonstrate the importance of epigenetic modifications in regulating *Gata4* functions (Figure 3) (and those of other factors) in cardiogenesis confirm other findings. In fact, work by Si et al. suggests that the *Smad4*-mediated *Bmp2* signalling pathway affects the histone H3 acetylation of *Gata4* and *Nkx2.5* genomic loci, which is essential for their regulation [65]. Those data are in line with other reports on system biology involving other cardiac factors [143]. The above-mentioned data not only demonstrate the importance of epigenetic modifications of the *Gata4* locus during cardiac development, but they also suggest the crucial role played by those post-translational modifications in heart development (Figure 3). Indeed, other investigations demonstrate that GATA4 binds and participates in establishing an active chromatin region by stimulating the H3K27ac deposit, which facilitates *Gata4*-driven gene expression [144]. 

Altogether, the data presented above clearly demonstrate the crucial role of *Gata4* (among the cardiogenic GATA factors) during heart formation. As described for *gata5* and *6* above, we also performed genome-wide analyses for the identification of specific targets for *gata4* during cardiomyogenesis [24,25]. Our work shows that *sox7* and *sox18*, whose function in cardiomyogenesis has previously been described [145], are specifically regulated by *gata4* in both Xenopus and mammalian systems (ES cells) [24,25] (Figure 1B). These studies also identified a wealth of target genes specifically regulated by *gata4* that pave the way for further investigations into elucidating GRNs that are involved downstream of *gata4* [25] during cardiomyogenesis. Figure 4 is a proposed model to depict *GATA4*, *5* and *6* functions during heart formation.

## 7. Conclusions and Further Perspectives

In summary, studies from recent years have not only provided a wealth of evidence supporting the importance of *Gata4*, *5* and *6* in cardiogenesis and heart diseases but also molecular mechanisms of their activities and the identification of their target genes. For instance, a strong role of *Gata5* in endocardium formation and its involvement in aortic valve formation has been demonstrated. However, a level of complexity with regards to the function of these factors in heart formation has also been uncovered with the discovery in mice of the interactions among *Gata5* and its other sisters *Gata4* and *6* in outflow tract (OFT) formation. Consistent with that, compelling evidence for an important role of *Gata4* in heart septation and valve formation has been provided. Furthermore, molecular mechanisms involved in the latter have demonstrated that *Gata4* interacts with other partners, such as chromatin remodelling factors, resulting in either acetylation or deacetylation that drives proper gene expression in the atrioventricular channel and ventricular chambers. Those data once more highlight the importance of interacting partners and are consistent with the observation that the cardiac malformation seen in humans, when *GATA4* is disrupted, is similar to those seen with Holt–Oram syndrome mutations in *TBX5* and when *NKX2.5* is mutated. Although many other studies have also uncovered other interacting partners of GATA4 beyond those mentioned in this review, it is worth speculating that many others will be discovered in the future not only for GATA4 but also for its other sisters GATA5 and GATA6. Interacting with other partners allows GATA factors to regulate cardiogenesis through signalling pathways, some of which have been discovered in recent years, with likely more to come. The recent identification of targets specifically regulated by each of these factors during cardiogenesis opens the way to studies that will allow elucidating not only for which part of the heart those factors are required but also the molecular mechanisms involved.

## Figures and Tables

**Figure 1 ijms-23-05255-f001:**
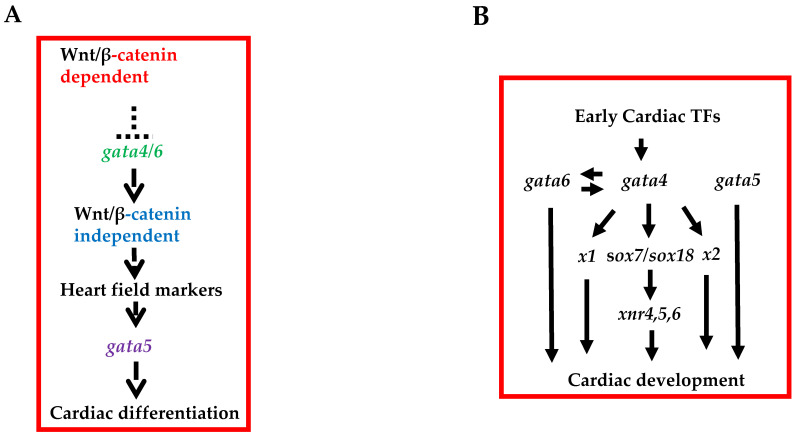
Schematic to depict the functional differences between cardiogenic factors during heart formation. (**A**) β-catenin-dependent Wnt signalling represses *gata4/6*, which in turn activates noncanonical Wnt signalling for induction of heart field markers after which *gata5* is induced to promote cardiac differentiation. (**B**) All cardiogenic GATA factors function after induction of early cardiogenic transcription factors where *sox7* and *sox18* act downstream to mediate only some of *gata4*’s function, not those of *gata5* or *gata6* during cardiomyogenesis. Abbreviations: TFs, Transcription factors. X1 and x2 indicate yet unidentified intermediary factors. (Panel B as in [24]).

**Figure 2 ijms-23-05255-f002:**
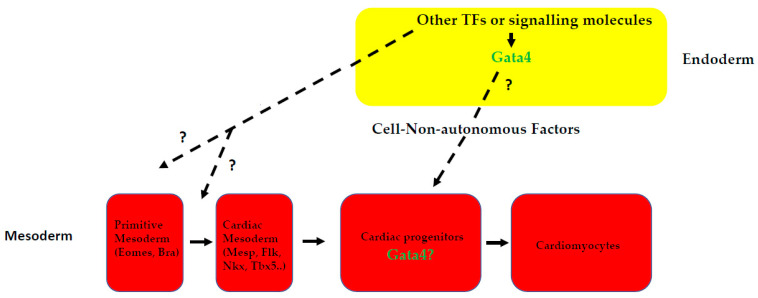
Factors emanating from endoderm induce mesoderm layer formation. Gata4 in endoderm controls a non-cell-autonomous pathway for cardiac myogenesis. Other transcription factors and signalling molecules induce Gata4 in endoderm, which in turn induces secreted molecules that promote cardiac differentiation of the underlying cardiac progenitors to cardiomyocytes. This model did not exclude a direct function of Gata4 in the mesoderm itself to promote cardiac differentiation as depicted by the question mark (?). Abbreviations: Eomes, Eomesodermin; Bra, Brachyury; Mesp, Mesoderm Posterior; Nkx, Nkx gene; Tbx, Tbx gene; Gata4, Gata4 gene.

**Figure 3 ijms-23-05255-f003:**
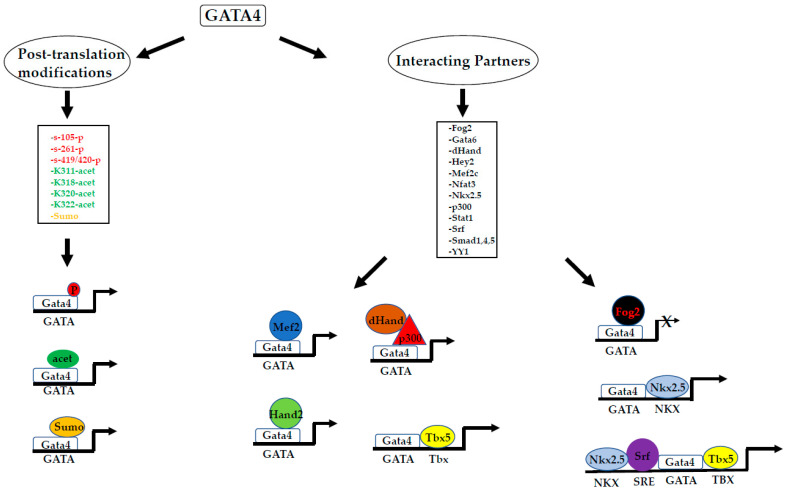
Schematic of cardiac transcriptional activity regulated by Gata4 with several other transcription factors showing that post-translational modifications of Gata4 is at the centre of Gene Regulatory Networks (GRNs) regulating cardiac formation and function. Abbreviations: P, Phosphorylation; Acet, Acetylation; Sumo, SUMOylation.

**Figure 4 ijms-23-05255-f004:**
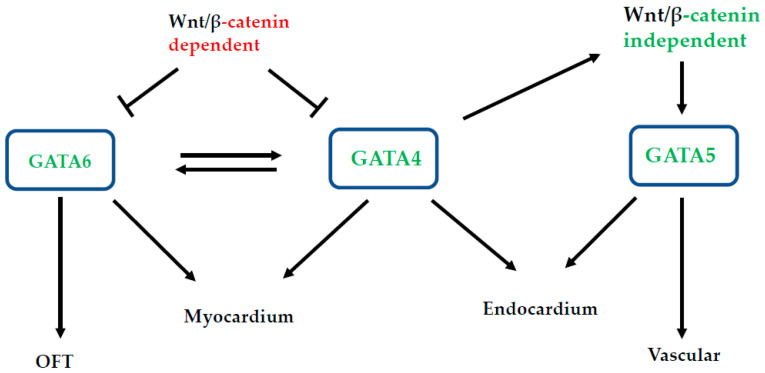
Proposed summary of Gata functions in heart development.

**Table 1 ijms-23-05255-t001:** The importance of Gata4, 5 and 6 in heart formation. Biological functions and cardiac defects associated in humans and mice.

GATA Factors	Cardiac Defects Associated with Human Mutations	Mouse Phenotypes	Cardiac Expression	Biological Functions
*GATA6*/*Gata6*	-Atrial septal defects -Atrioventricular septal defects -Bicuspid aortic valves -Dilated cardiomyopathy -Double outlet left ventricle -Double outlet right ventricle -Tetralogy of Fallot -Ventricular septal defects	Embryonic lethal at E5.5–E7	Yes	-Cardiac hypertrophy [13] -Cardiac morphogenesis [14] -Outflow tract development [15,16,17,18]
*GATA5*/*Gata5*	-Atrial septal defects -Atrioventricular septal defects -Bicuspid aortic valves -Dilated cardiomyopathy -Double outlet right ventricle -Hypertrophic cardiomyopathy -Tetralogy of Fallot -Ventricular septal defects	Viable and fertile	Yes	-Endocardial cell development [19] -Vascular endothelial homeostasis [20]
*GATA4*/*Gata4*	-Atrial septal defects -Atrioventricular septal defects -Bicuspid aortic valves -Dilated cardiomyopathy -Double inlet left ventricle -Double outlet right ventricle -Hypertrophic cardiomyopathy -Tetralogy of Fallot -Ventricular septal defects	Embryonic lethal at E9.5	Yes	-Cardiomyocyte proliferation, differentiation and hypertrophy [13] -Endocardial cell proliferation [21] -Heart tube formation and ventral morphogenesis [22,23]

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
