# Peer review of "Towards Understanding the Gene-Specific Roles of GATA Factors in Heart Development: Does GATA4 Lead the Way?"

_ijms, 2022, doi:10.3390/ijms23095255_

Round 1

Reviewer 1 Report

This is a review article so obviously there is no novelty to the work, which is acceptable in this case. However, the major problem with this work is that there are absolutely no figures in the paper. It is just walls of text with no breaks! A few color figures are needed for the readers to understand the role of GATA factors in heart development. I would say at least 3-4 figures are needed to make this manuscript more accessible to readers.

Author Response

Response to Reviewer 1 Comments

Point 1: “This is a review article so obviously there is no novelty to the work, which is acceptable in this case. However, the major problem with this work is that there are absolutely no figures in the paper. It is just walls of text with no breaks! A few color figures are needed for the readers to understand the role of GATA factors in heart development. I would say at least 3-4 figures are needed to make this manuscript more accessible to readers”.

Response 1: I appreciate the reviewer’s comments about the manuscript. As requested by the reviewer the manuscript has now coloured figures which I believe will help readers to understand the role of the described factors in heart development and diseases.

Reviewer 2 Report

The authors introduced the importance of Gata4, 5 and 6 in cardiogenesis and heart diseases and the molecular mechanisms of actions. This would provide a great overview of the existing literature on this topic. The reviewer has a few comments which the authors may address:

  1. As IJMS only accepts comprehensive reviews, the authors should update the references (e.g., 100+ and 50% are publications in the last five years).
  2. The author must select a title that grabs attention, accurately describes the contents of the manuscript, and makes researchers want to read further. There is no direct evidence to lead GATA4 only. The word ‘Devel-Opment’ should be corrected.
  3. Novelty and rationale of the study: The author should clearly focus on fixed ideas, use a procedural and critical approach to the literature and express the findings in an attractive way.
  4. Previous studies have underscored the pivotal roles of the cardiac transcription factors in cardiac morphogenesis and the proliferation, specification, and differentiation of cardiomyocytes, including the GATA zinc finger-containing transcription factor. A great number of mutations in GATA binding protein (GATA)4, GATA5, GATA6 have shown to be involved in various congenital diseases. Do they have mutations in cardiac development and how could they affect this process?
  5. What is the correlation between Gata4 and Gata6 in cardiac and vascular development? The author should critically analyze the previously published studies, point out weaknesses in those studies and suggest future courses of action.
  6. Schematic diagrams of Gata4, 5 and 6 interactions are needed to help readers better understand their function in cardiac development. At least two summative figures are needed. The author should draw upon the recent articles that they review to suggest new research directions, to strengthen support for existing theories and/or identify patterns among existing research studies.
  7. The direct and indirect role of Gata4 and Gata5 in cardiac development should be discussed in depth. The author should summarize the arguments and ideas of others without adding one’s own contributions.

Author Response

Response to Reviewer 2 Comments

I appreciate the reviewer acknowledges my introduction of the importance of Gata4, 5 and 6 in cardiogenesis and heart diseases and the molecular mechanisms of their actions. Thus, the current manuscript is more needed and would provide a good overview of the existing literature. However, the reviewer provided a few comments, which I have addressed, as detailed below.

Point 1: “As IJMS only accepts comprehensive reviews, the authors should update the references (e.g., 100+ and 50% are publications in the last five years)”.

Response 1: As requested, the references are updated with more than 100 references with a heathty amount of more recent references. Thus, the manuscript in its present version now goes well beyond its initial scope/aim by providing and integrating those newly added references.

Point 2:  “The author must select a title that grabs attention, accurately describes the contents of the manuscript, and makes researchers want to read further. There is no direct evidence to lead GATA4 only.

Response 2: I thought my title was grabbing attention. But I did make a change with a new title “Towards understanding the Gene-Specific Roles of GATA Factors in Heart Development: Does GATA4 Lead the Way?”

Point 2: “There is no direct evidence to lead GATA4 only.”

Response 2: It is unclear what exactly the reviewer is arguing for here. The provided figures and extended text show the importance of Gata4, but of course in the context of the other cardiogenic GATA transcription factors.

Point 2: “The word ‘Devel-Opment’ should be corrected.”

Response 2: This was a typographic error which somehow was introduced after submission. It has now been corrected. Hopefully this will not happen again after resubmission.

Point 3: “Novelty and rationale of the study: The author should clearly focus on fixed ideas, use a procedural and critical approach to the literature and express the findings in an attractive way.”

Response 3: I have tried to highlight new findings in the field in line with my initial scope/aim of the manuscript. But as mentioned above, I did add and incorporate more references in the revised manuscript which went well beyond my initial aim to accommodate the reviewer comments. It is not clear what the reviewer was trying to say or where exactly the reviewer was suggesting adding more focus on fixed ideas.

Point 4: “ Previous studies have underscored the pivotal roles of the cardiac transcription factors in cardiac morphogenesis and the proliferation, specification, and differentiation of cardiomyocytes, including the GATA zinc finger-containing transcription factor. A great number of mutations in GATA binding protein (GATA)4, GATA5, GATA6 have shown to be involved in various congenital diseases. Do they have mutations in cardiac development and how could they affect this process?

Response 4: This review was not meant to be focusing on congenital diseases. But I have mentioned that where appropriate in sections 2, 3 and other sections in the manuscript, for instance when specific points needed to be made.  I believe that providing details on how Gata4, 5 and 6 affect congenital heart disease as requested by the reviewer would require writing an entire book, which obviously goes well beyond the scope of the current manuscript.

Point 5: “What is the correlation between Gata4 and Gata6 in cardiac and vascular development? The author should critically analyse the previously published studies, point out weaknesses in those studies and suggest future courses of action.”

Response 5:  Again, addressing this point would require writing another review. I did expatiate on the genetic and molecular interaction between all cardiogenic Gata (Gata4, 5 and 6) in this review (see section 2 and other sections when detailing individual function of each factor). I was not going to be writing a review about vascular development. Adequately addressing the point made by the reviewer here would again require writing a dedicated review on the role of these factors in vascular development and disease.

Point 6: “Schematic diagrams of Gata4, 5 and 6 interactions are needed to help readers better understand their function in cardiac development. At least two summative figures are needed. The author should draw upon the recent articles that they review to suggest new research directions, to strengthen support for existing theories and/or identify patterns among existing research studies.”

Response 6: I appreciate the reviewer point to summarise finding in diagrams and figures as suggested by another reviewer. Schematic diagrams have now been added. I believe those newly added diagrams will greatly help readers understand the roles of Gata4, 5 and 6 in cardiogenesis and heart diseases and the molecular mechanisms of their actions.

Point 7: “The direct and indirect role of Gata4 and Gata5 in cardiac development should be discussed in depth. The author should summarize the arguments and ideas of others without adding one’s own contributions.”

Response 7: One of my aims in writing this review was precisely to place my own contribution in the context of wide literature in the heart development research field, especially the role of those transcription factors.

Round 2

Reviewer 1 Report

The author has made the requested revisions.

Reviewer 2 Report

In the revised article, the authors modified the manuscript referred to the comments, and answered the questions comprehensively.